

# Anodal transcranial direct current stimulation enhances response inhibition and attention allocation in fencers

Jiansong Dai[1,*], Yang Xiao[2,*], Gangrui Chen[3], Zhongke Gu[1] and Kai Xu[1]

[1] Department of Sport and Health Sciences, Nanjing Sport Institute, Nanjing, China
[2] Department of Graduate, Nanjing Sport Institute, Nanjing, China
[3] Department of Sport Research, Nanjing Sport Institute, Nanjing, China
[*] These authors contributed equally to this work.

## ABSTRACT

**Background**. The aim of this study is to investigate the acute effects of anodal transcranial direct current stimulation (tDCS) on reaction time, response inhibition and attention in fencers.

**Methods**. Sixteen professional female fencers were recruited, and subjected to anodal tDCS and sham stimulation in the primary motor area (M1) one week apart in a randomized, crossover, single-blind design. A two-factor analysis of variance with repeated measures was used to analyze the effects of stimulation conditions (anodal stimulation, sham stimulation) and time (pre-stimulation, post-stimulation) on reaction time, response inhibition, and attention in fencers.

**Results**. The study found a significant improvement in response inhibition and attention allocation from pre-stimulation to post-stimulation following anodal tDCS but not after sham stimulation. There was no statistically significant improvement in reaction time and selective attention.

**Conclusions**. A single session of anodal tDCS could improve response inhibition, attention allocation in female fencers. This shows that tDCS has potential to improve aspects of an athlete's cognitive performance, although we do not know if such improvements would transfer to improved performance in competition. However, more studies involving all genders, large samples, and different sports groups are needed in the future to further validate the effect of tDCS in improving the cognitive performance of athletes.

## INTRODUCTION

Fencing is a one-on-one combat sport that involves close-range attacks and very fast exchanges of attack and defense. The sport requires fencers to maintain a high level of attention during the short time between each attack and defense move, to judge the spatial position and distance between the opponent, the foil and themselves, and always be ready to change and suppress the action in a very short time, which places high demands on the fencer's speed of information processing, response inhibition and attention (*Doğan, 2009*;

Corresponding author
Jiansong Dai, daijiansong@163.com

*Gutiérrez-Davila et al., 2019*; *Hijazi, 2013*). Reaction time (RT) is a widely used measure of information processing speed. It is defined as the time elapsed between the onset of a stimulation (*e.g.*, visual or auditory) and an individual's response (*Kranzler, 2012*). Previous studies have established that rapid stimulation recognition and response, as well as response inhibition, are crucial factors for successful fencing performance (*Borysiuk, 2008*; *Zhang et al., 2015*). Response inhibition is the ability to inhibit inappropriate or unrequired behaviors to allow flexible and goal-driven behavioral responses to environmental changes, which is an important component of executive function (*Diamond, 2013*; *Sandrini et al., 2020*). According to *Di Russo et al. (2006)* and *Chan et al. (2011)*, expert fencers demonstrate higher response inhibition rates compared to novices. Additionally, athletes perform better than non-athletes in this cognitive domain. Response inhibition can also be utilized to evaluate athletic potential (*Brevers et al., 2018*; *Di Russo et al., 2006*; *Zhang et al., 2015*). Attention is a complex cognitive function that involves several components, including the alerting, orienting, and executive systems that guide mental activity to focus on a given task (*Petersen & Posner, 2012*; *Roy et al., 2015*). It is a key determinant of fencing outcomes in competitive settings, as evidenced by a significant correlation between attentional measures and fencing scores (*Gutiérrez-Davila et al., 2017*; *Hijazi, 2013*). The above information highlights the significance of reaction time, response inhibition, and attention in fencing. These cognitive functions enable fencers to maintain focus, react quickly, and have better control over their responses to false moves during a match, ultimately increasing the likelihood of winning. However, it is unclear whether fencers can improve these aspects of cognitive function through non-training stimulation.

Transcranial direct current stimulation (tDCS) is a non-invasive brain stimulation technique that uses constant weak currents to modulate neuronal excitability in specific areas of the cerebral cortex. The basic mechanism of neuronal excitability modulation is to alter the resting membrane potential of neurons in the stimulated brain region. Anodal tDCS (a-tDCS) depolarizes the neuronal membrane potential and increases neuronal excitability, and cathodal tDCS (c-tDCS) hyperpolarizes the neuronal membrane potential and decreases neuronal excitability (*George & Aston-Jones, 2010*; *Nitsche et al., 2008*; *Notturno et al., 2014*). Previous studies have indicated that a-tDCS applied to supplementary motor area (SMA), superior medial frontal cortex (SMFC), and primary motor cortex (M1) can reduce reaction time in healthy people (*Bender, Filmer & Dux, 2017*; *Carlsen, Eagles & MacKinnon, 2015*; *Molero-Chamizo et al., 2018*). And several studies have confirmed that a-tDCS applied to M1 and pre-supplementary motor area (pre-SMA) enhances response inhibition in healthy populations (*Kuo et al., 2013*; *Liang et al., 2014*; *Yu et al., 2021*). In addition a-tDCS applied to dorsolateral prefrontal cortex (DLPFC) has been shown to improve attentional function in healthy young adults (*Fukai et al., 2019*; *Lu et al., 2020*). In the aforementioned preliminary study (*Bender, Filmer & Dux, 2017*; *Fukai et al., 2019*; *Liang et al., 2014*; *Lu et al., 2020*; *Molero-Chamizo et al., 2018*; *Yu et al., 2021*), notable enhancements in reaction time, response inhibition, and attention were observed as a result of positive stimulation effects induced by tDCS. The studies mentioned above applied tDCS to various brain regions. However, there are fewer studies related to M1, we still have much to learn about how the effects of tDCS applied to this brain

regions influences aspects of cognitive function. In addition, the effect of tDCS on cognitive function in athletes is still much to be discovered.

M1 is located in the precentral gyrus, a key region involved in executing movement. A number of studies have examined the effects of M1 stimulation on reaction time, but few studies have examined the effects of M1 stimulation on response inhibition and attention. M1 receives and synthesizes multiple types of input from a variety of cortical and subcortical regions as a brain hub (*Dum & Strick, 2005*; *He, Dum & Strick, 1995*; *Muakkassa & Strick, 1979*). M1 is considered a key cortical site of response inhibition, mediating movement preparation and inhibition through M1's intrinsic motor programming circuitry, because of M1's role in encoding downward motor commands (*Bhattacharjee et al., 2021*; *Stinear, Coxon & Byblow, 2009*). Thus, M1 plays a critical role in shaping and encoding descending motor output. It also has an modulatory effect during response inhibition (*Alexander, De Long & Strick, 1986*; *Nambu, Tokuno & Takada, 2002*; *Stinear, Coxon & Byblow, 2009*). Additionally, there is evidence that tDCS can stimulate a wide range of brain activity (*Fiori et al., 2018*; *Meinzer et al., 2013*; *Weber et al., 2014*; *Zheng, Alsop & Schlaug, 2011*), potentially including areas that are crucial for attentional processing, leading to improved attention. Therefore, M1 stimulation is a novel and well-founded attempt to modify response inhibition and attention.

In summary, considering the established efficacy of tDCS in enhancing reaction time, response inhibition, and attention, and the dearth of research on its effects specifically on the M1 region, research is needed to examine the acute effects of tDCS applied to M1 on reaction time, response inhibition, and attention in fencers. The primary objective of this study was to examine the potential of a-tDCS to enhance reaction time, response inhibition, and attention in fencers, with the ultimate goal of helping them win competitions. We formulated the hypothesis that a-tDCS would lead to improvements in reaction time, response inhibition, attentional allocation, dynamic attention and selective attention among fencers.

## MATERIALS & METHODS

### Participants

This study enrolled 16 female fencers (age: 21.18 ± 1.32 years, height: 173.50 ± 5.81 cm, weight: 63.62 ± 7.73 kg, training years: 9.75 ± 1.18) from the Jiangsu Provincial Fencing Team of Nanjing Sports Institute. All subjects were right-handed, had good overall health, no history of mental illness or brain injury, were not allergic to direct current and volunteered to participate in the trial. G*power program was used to compute a priori sample size with a test family = F-tests, statistical test = analysis of variance (ANOVA): repeated measures between factors. Based on previous studies (*Codella et al., 2021*; *Maeda et al., 2017*), power, alpha, and effect size were set at 0.80, 0.05, and 0.4, respectively, calculating that 14 subjects were required for a single group. To allow for an anticipated dropout rate of 10%, there were 16 subjects for a single group. The subjects were informed of the basic test procedure and potential risks before the trial, and all signed an informed consent form. The study protocol was approved by the Ethics Committee of the Nanjing

Sports Institute (Approval no. RT-2022-06) and was conducted in accordance with the tenets of the Declaration of Helsinki.

## Study design and procedures

All the enrolled fencers underwent a-tDCS and sham stimulation one week apart in a crossover, randomized, single-blind manner to investigate the effects of anodal and sham stimulation on reaction time, response inhibition, and attention (Fig. 1). To minimize the effects of circadian rhythm changes, subjects were tested at the same time of day. Prior to each test, subjects were instructed to abstain from alcohol, caffeine, theophylline, drugs or supplements, and to get a good night's sleep. A standard experimental procedure is described as follows: the subjects came to the laboratory in the Physical Fitness Centre and sat in silence for 5 min to regain their composure (resting phase). A Stop Signal Task (SST) was then started, this will take approximately five minutes. After the end of the SST test, a Neuro tracker (NT) test was conducted in three modes: core (four balls), dynamic (four balls), and selective (two balls), with a 2-min interval between each test, each test takes approximately six minutes. The athletes were given five attempts to familiarize themselves with the test in each mode before starting the actual test. After completion of the whole test, an anodal tDCS intervention was carried out with a current strength of 2 mA for 20 min or sham stimulation (*Bashir et al., 2022*; *Gu et al., 2022*). Another SST test was performed immediately after the stimulation. After the end of the SST test, NT test was conducted in three modes: core, dynamic, and selective, with a 2-min interval between each test. However, the NT test in each mode was directly tested without attempts.

## tDCS intervention

The tDCS intervention was carried out using the 1300A&4x1-C3A multichannel tDCS apparatus (Soterix Medical, New York, NY, USA). The 16 athletes were randomly assigned numbers, and each athlete had to undergo both anodal and sham stimulation. A crossover randomized design was used to determine the condition of stimulation for the first round of 16 athletes, followed by the second round of stimulation in the opposite condition, 1 week later. tDCS was administered by personnel familiar with the operation of the instrument, but they were not involved in this study, and the subjects were blinded to the type of stimulation received. For the safety of the subjects, the procedures related to the use of tDCS were designed based on the study by *Muthalib et al. (2018)*. The left M1 primary motor area was selected and electrodes were placed on C3, FC1, FC5, CP5, and CP1 based on the International 10-10 system, with C3 being the central electrode (Fig. 2). Five silver chloride sintered circular electrodes (one $cm^2$) were connected to a 4 x 1 adapter; the adapter was then attached to the tDCS device using a cable, and the electrode fixation caps were attached to the electrode caps at C3, FC1, FC5, CP5, and CP1.

The subjects were asked to sit quietly on a back-rest chair. An experimenter then left the subject's scalp exposed, injected conductive gel into the five electrode fixation caps, and positioned the electrodes in place. When anodal stimulation was performed, the operator pressed the conductivity test button to check the conductivity and then pressed the start button to perform tDCS. The current strength was gradually increased to 2 mA within

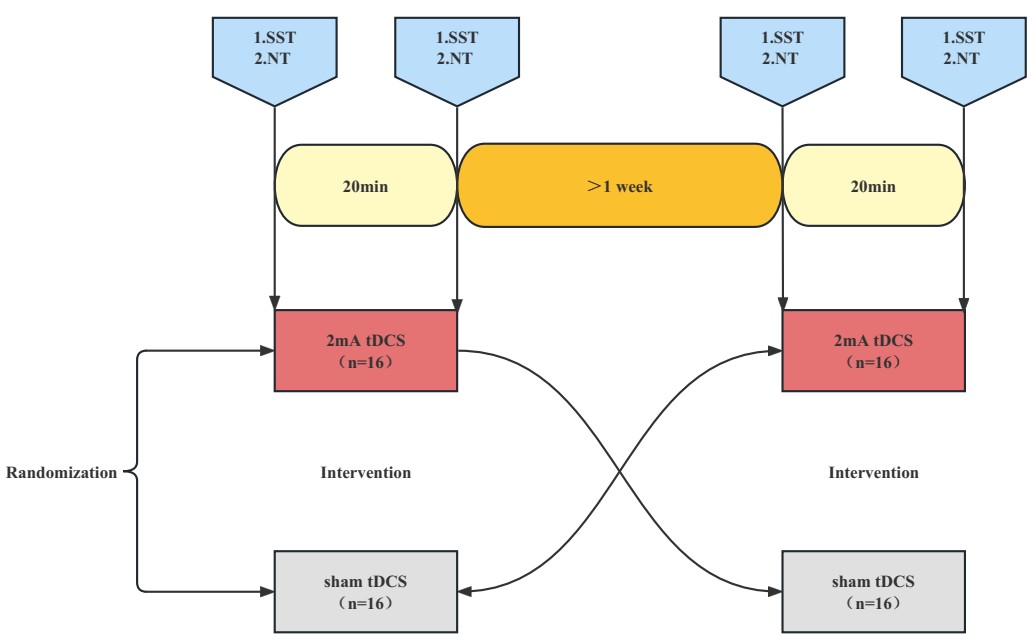

**Figure 1 Trial flow chart.** SST, stop signal task. NT, Neuro tracker test. Subjects were randomized to anodal tDCS or sham tDCS and tested for SST and NT before and after stimulation. After an interval of one week, the anodal tDCS subjects were crossed over with the sham tDCS subjects and were tested for SST and NT before and after stimulation.

30 s and lasted 20 min, and then the current gradually decreased to 0 mA in the last 30 s. The current intensity, stimulation time, and current acceleration of the sham stimulation were the same as those of the anodal stimulation, except that the current accelerated to 2 mA in the first 30 s and then gradually decreased to 0 mA automatically in 30 s (*Gu et al., 2022*; *Guo et al., 2022*). The tDCS was stopped immediately if the subject felt any discomfort other than the normal mild tingling sensation of stimulation throughout the tDCS procedure.

## Data collection
### SST test

The SST program was written using the E-Prime 3.0 software (Psychology Software Tools, Inc., Sharpsburg, PA, USA), it is widely used to test the subjects' response inhibition ability (*Guo et al., 2022*; *Sandrini et al., 2020*). The Go task reaction time (Go RT) and Stop-signal reaction time (SSRT) were selected as indicators of reaction time and response inhibition. The GO RT was determined by measuring the time interval between the time when go signal appeared and time when participants made the correct response. A total of 240 trials were performed, including 180 (75%) Go trials, 20 (8.3%) NoGo trials, and 40 (16.7%) stop-signal trials. Trials were presented randomly, and the number and type of tasks were equally divided between the two blocks, with an intergroup interval of no more than 2 min. Prior to the start of the experiment, the subjects were informed of the testing procedure and proceeded with 25 trial attempts until they were familiar with the task, fully

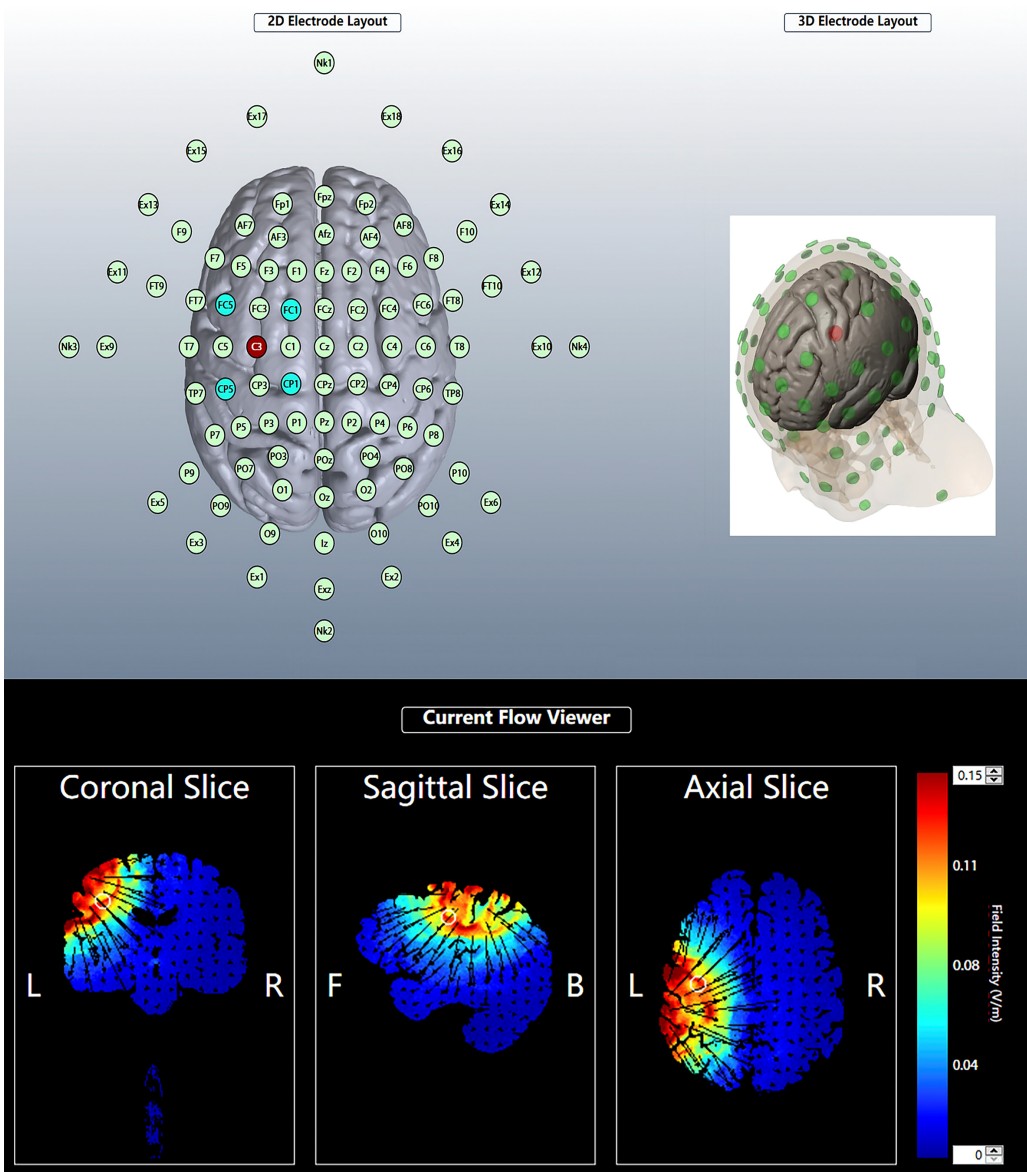

**Figure 2 Electrode position and current distribution.** The red dots in the 3D and 2D electrode layout represent the center (anodal) electrode. The center electrode was placed at C3. The four cathodal electrodes were placed on FC5, CP5, FC1 and CP1. For each plane, the theoretical current is estimated by the Soterix HD –Explore based on the modelled electric field normal component (nE, V/m) at the cortex of the high precision DC stimulation electrode array. Figures are color-coded according to the electric field strength, Blue indicates zero electric field (0 V/m) and red indicates peak magnitude (0.15 V/m).

understood the task, and became proficient in its operation. In the Go trial, the central position was initially calibrated with a cross on the computer screen, and the subjects used the keyboard to press the appropriate left and right keys according to the left and right black arrows that appeared in the center of the computer screen. The arrows were presented for 1,000 ms. The next trial would appear after the response (regardless of its accuracy) or

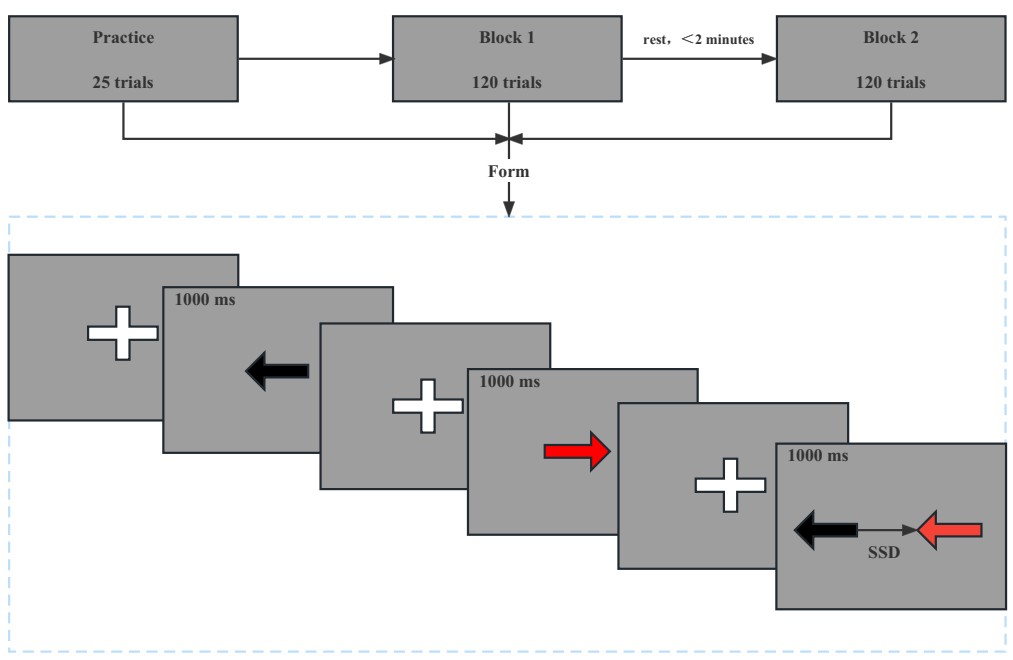

**Figure 3 Stop signal task mode.** In the Go trial, a cross appears in the center of the computer screen and then a left or right black arrow appears for 1,000 ms, as shown in the first two panels. In the NoGo trial, a cross appears in the center of the computer screen and then a left or right red arrow appears for 1,000 ms, as shown in the third and fourth panels. In the SST trial, a cross appears in the center of the computer screen and then a left or right black arrows appeared on the screen initially, but after the stop-signal delay, the arrows changed color to red, as shown in the last two panels.

if no response was recorded within 1,000 ms. In the SST trial, the left/right black arrows appeared on the screen initially, but after the stop-signal delay (SSD), the arrows changed color to red, to which the subject is to inhibit their response and did not respond. The initial value of SSD was 250 ms, and a 50% success suppression was maintained by a 50-ms incremental correct/50-ms decremental incorrect algorithm, which has been widely used in stop-signal studies (*Friehs & Frings, 2019*; *Huijgen et al., 2015*; *Parkin & Walsh, 2017*). The SSRT calculation was simplified as follows: mean SSRT = mean Go RT - mean SSD. SSRT represents an estimate of how long it takes a person to successfully inhibit their prepotent response and that a shorter SSRT indicates better response inhibition. In the NoGo trial, A red arrow appears on the screen but the subjects did not respond to the red arrow (1,000 ms) (Fig. 3).

## NT test

Subjects' attention was tested using the Neuro Tracker multi-task tracking neuro-perceptual system (Faubert Applied Research Centre, Toronto, Canada). To increase stereoscopy and the demands of attentional tracking, the NT test involves tracking multiple targets among distractors in a 3D environment. Thus, NT involves multifocal attention and requires a high level of processing that includes the brain's ability to ignore distractors and maintain focus (*Cavanagh & Alvarez, 2005*; *Meyerhoff, Papenmeier & Huff, 2017*). NT is based on a

multi-target tracking task (MOT), Originating with *Pylyshyn & Storm (1988)*, MOT is a well-established paradigm for studying attention. The MOT task has been used to assess sustained, distributed, dynamic, and selective attention and has proven to be one of the most useful and popular tools in attention research (*Doran and Hoffman, 2010*; *Drew et al., 2009*; *Koldewyn et al., 2013*; *Meyerhoff, Papenmeier & Huff, 2017*; *Meyerhoff & Papenmeier, 2020*). In the NT test, speed threshold scores measured in the core, dynamic, and selective modes were selected as a measure of attention. The NT test was conducted sequentially in three modes: core, dynamic, and selective. Subjects were tested once for each mode, and 20 trials were performed each time. In the core mode, the subject was seated in a back-rest chair with 3D glasses, and her line of sight was aligned with the center of the cube. Eight small yellow-green static balls appeared in the cube's field of view. The four target balls to be tracked by the subject appeared in red for 2 s and then turned back to the original yellow-green color. The eight balls then began to move and collide at their initial velocity in an irregular pattern in the cube's field of view, changing direction of motion when they collided with each other or with the walls. After 8 s, the balls stopped moving and were marked from 1 to 8. The subject was required to say the number corresponding to the target ball and was judged to be correct only if all the numbers of the target balls were reported correctly. The correct target balls were shown in red for 2 s after the subject reported the number, after which the next trial began.

After each trial, the ball speed automatically adjusted based on the correctness of the responses, with correct answers increasing the speed of the ball and incorrect answers decreasing it. Thus, the correct rate was controlled at around 50%. At the end of the test, the subject was given a speed threshold score as a test indicator. Dynamic mode differed from the core mode in that the sphere moved in an irregular manner at variable speed. In the dynamic mode, the ball speed slowed down and then returned to the initial speed over the course of 8 s, with the remaining procedure identical to the core mode. Selective mode differed from the core mode in that there were only two target balls, one blue and one red ball, and at the end of the trial, the subject reported the blue ball first and then the red ball. The rest of the procedure was the same as that in the core mode.

## Statistical analysis

All data were statistically analyzed using JMP, version 16 (SAS Institute, Inc., Cary, NC, USA) and index data were expressed as the mean standard deviation (M ± SD). A two-factor analysis of variance (ANOVA) with repeated measures was used to analyze the effects of stimulation condition (anodal and sham stimuli) and time (pre-stimulation and post-stimulation) on reaction time (Go RT), response inhibition (SSRT), and attention ability (speed threshold scores). *Post-hoc* analysis with LSD was used if significant main or interaction effects were observed. A value of $P < 0.05$ was considered to indicate significant difference. The effect size (ES) analysis was used to report the magnitude of the pre-post differences for all measures in each condition, and the ES was expressed as Cohen's d. The criteria for interpreting ES size were: <0.19 for a weak effect, 0.20–0.49 for a low effect, 0.50–0.79 for a medium effect, and >0.8 for a high effect (*Cohen, 1992*).

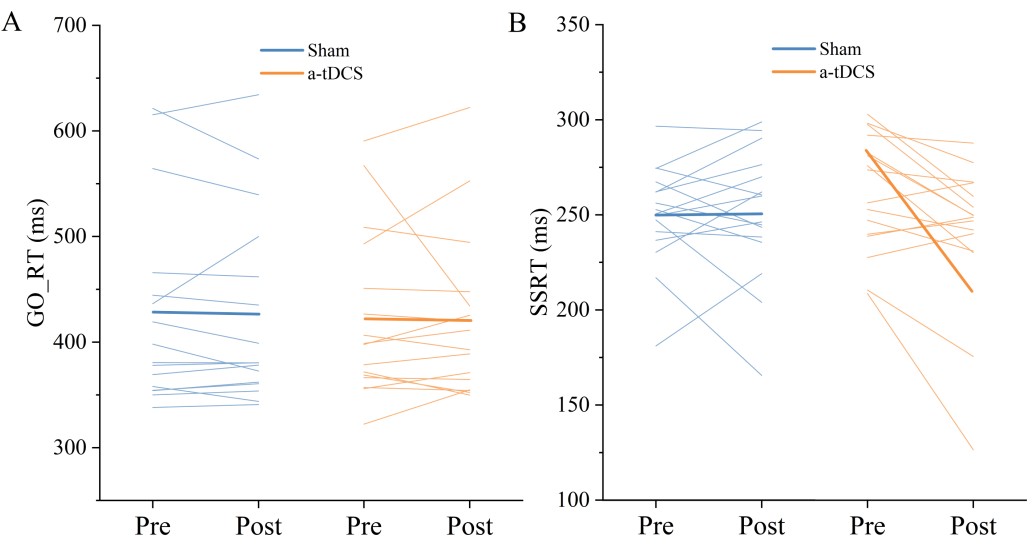

**Figure 4** **Trends of GO_RT (A) and SSRT (B) before (pre) and after (post) stimulation in each subject after tDCS (a-tDCS and sham).** Bold lines represent mean data.

## RESULTS

### Go RT

Two-factor repeated measures ANOVA was used to analyze the effect of Stimulation conditions and time (pre-stimulation and post-stimulation) on reaction time. The results indicated that the main effect of stimulation conditions [$F_{(1,30)} = 0.030$, $P = 0.862$, $\eta_p^2 = 0.001$] and main effect of time [$F_{(1,30)} = 0.083$, $P = 0.774$, $\eta_p^2 = 0.003$] were both not significant. The interaction between time and stimulation conditions was insignificant [$F_{(1,30)} = 0.001$, $P = 0.974$, $\eta_p^2 < 0.001$]. *Post-hoc* analysis showed no significant differences in Go RT from pre to post stimulation for anodal or sham stimulation (Fig. 4).

### SSRT

Two-factor repeated measures ANOVA was used to analyze the effect of Stimulation conditions and time (pre-stimulation and post-stimulation) on SSRT. The results showed that the main effect of stimulation conditions was not significant [$F_{(1, 30)} = 0.008$, $P = 0.928$, $\eta_p^2 < 0.001$], while the main effect of time was significant [$F_{(1,30)} = 4.711$, $P = 0.038$, $\eta_p^2 = 0.135$]. The interaction between time and stimulation conditions was significant [$F_{(1, 30)} = 5.299$, $P = 0.028$, $\eta_p^2 = 0.150$]. Results of *post-hoc* analysis demonstrated that SSRT was significantly reduced after a-tDCS ($P < 0.05$) (Table 1, Fig. 5).

### Speed threshold scores

A two-factor repeated measures ANOVA was used for the analysis of the speed threshold score in the core mode. We found that a main effect of stimulation condition was not significant [$F_{(1,30)} = 0.134$, $P = 0.716$, $\eta_p^2 = 0.004$], a main effect of time was significant [$F_{(1,30)} = 9.764$, $P = 0.003$, $\eta_p^2 = 0.246$], and an interaction between time and stimulation

**Table 1  Effect of a-tDCS on reaction time and response inhibition.**

| Outcome | Sham | | | | a-tDCS | | | |
|---|---|---|---|---|---|---|---|---|
| | Pre | Post | ES | Classification | Pre | Post | ES | Classification |
| GO_RT (ms) | 427.91 ± 93.81 | 425.98 ± 90.37 | 0.02 | Trivial | 422.54 ± 78.80 | 421.00 ± 77.61 | 0.02 | Trivial |
| SSRT (ms) | 249.97 ± 26.46 | 250.58 ± 34.68 | 0.02 | Trivial | 261.63 ± 30.88 | 240.89 ± 39.37* | 0.57 | Moderate |

Notes.
a-tDCS, anodal transcranial direct current stimulation.
*Significant difference with Pre.

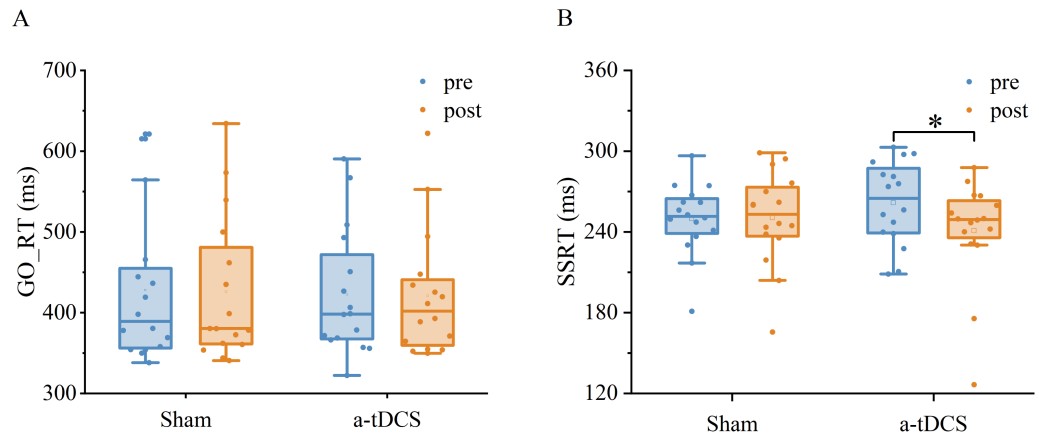

**Figure 5  Effects of tDCS on GO_RT(A) and SSRT(B).** Data are presented as M ± SD. Asterisks (∗) indicate statistically significant difference compared to pre-stimulation (∗P < 0.05).

**Table 2  Effect of a-tDCS on attention.**

| Outcome | Sham | | | | a-tDCS | | | |
|---|---|---|---|---|---|---|---|---|
| | Pre | Post | ES | Classification | Pre | Post | ES | Classification |
| Core (score) | 2.01 ± 0.57 | 2.09 ± 0.44 | 0.15 | Trivial | 1.77 ± 0.53 | 2.22 ± 0.36* | 0.88 | Large |
| Dynamic (score) | 1.88 ± 0.46 | 2.12 ± 0.44 | 0.51 | Moderate | 1.66 ± 0.43 | 1.98 ± 0.30* | 0.77 | Moderate |
| Selective (score) | 3.21 ± 0.64 | 3.50 ± 0.56 | 0.47 | Small | 2.98 ± 0.58 | 3.24 ± 0.58 | 0.43 | Small |

Notes.
a-tDCS, anodal transcranial direct current stimulation.
*Significant difference with Pre.

condition was significant [$F_{(1,30)} = 4.849$, $P = 0.035$, $\eta_p^2 = 0.139$]. Results of *post-hoc* analysis demonstrated that core mode speed threshold score increased significantly after a-tDCS ($P < 0.05$) (Table 2).

A two-factor repeated measures ANOVA was used for the analysis of the speed threshold score in the dynamic mode. We found no significant main effect of stimulation condition [$F_{(1,30)} = 2,140$, $P = 0.153$, $\eta_p^2 = 0.067$], a significant main effect of time [$F_{(1,30)} = 11,955$, $P = 0,001$, $\eta_p^2 = 0.285$] and an interaction between time and stimulation condition was not significant [$F_{(1, 30)} = 0.204$, $P = 0.654$, $\eta_p^2 = 0.007$]. Results of *post-hoc* analysis

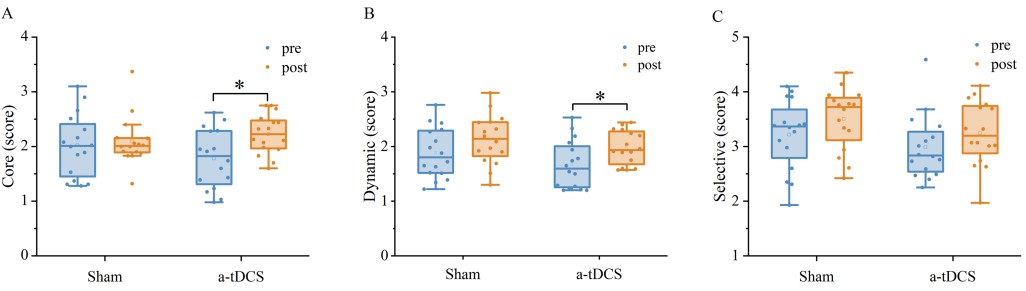

**Figure 6 Effects of tDCS on the speed threshold score of the core mode (A), dynamic mode (B) and selective mode (C).** Data are presented as M ± SD. Asterisks (∗) indicate statistically significant difference compared to pre-stimulation (∗P < 0.05).

demonstrated that dynamic mode speed threshold score increased significantly after a-tDCS ($p < 0.05$, Fig. 6).

A two-factor repeated measures ANOVA was used for the analysis of the speed threshold score in the selective mode. We found that a main effect of stimulation condition was not significant [$F_{(1,30)} = 2.963$, $P = 0.095$, $\eta_p^2 = 0.090$], a main effect of time was not significant [$F_{(1,30)} = 3.0985$, $P = 0.088$, $\eta_p^2 = 0.094$], and an interaction between time and stimulation condition was also not significant [$F_{(1,30)} = 0.007$, $P = 0.930$, $\eta_p^2 < 0.001$]. *Post-hoc* analysis showed no significant differences in selective mode speed threshold from pre to post stimulation for anodal or sham stimulation (Fig. 7).

## DISCUSSION

The purpose of this study was to investigate the acute effects of anodal transcranial direct current stimulation on reaction time, response inhibition, and attention in fencers. Result showed significant improvement in response inhibition and attention after a-tDCS. Previous literature has investigated the effects of tDCS on cognitive function and found positive results in reaction time (*Bender, Filmer & Dux, 2017*; *Carlsen, Eagles & MacKinnon, 2015*; *Molero-Chamizo et al., 2018*), response inhibition (*Kuo et al., 2013*; *Liang et al., 2014*; *Yu et al., 2021*) and attention allocation (*Fukai et al., 2019*; *Lu et al., 2020*); however, current studies on response inhibition and attention in the M1 brain region are scarce. In addition, subjects were all from the general population and studies with professional athletes are lacking. This study presents a novel approach by targeting tDCS intervention in M1 brain regions of fencers to investigate cognitive function. The results showed positive effects of tDCS on response inhibition and attention allocation, suggesting that tDCS may be a promising approach to improve cognitive performance in athletes.

The significance of reaction time in various sports cannot be overstated. Whether it is the simple response of sprinters and swimmers to the starting gun, or the complex response of fencing and boxing athletes to the opponent's instant attack and defense transition, all require the central nervous system to make rapid and accurate decisions and responses (*Vantorre, Chollet & Seifert, 2014*; *Kamali et al., 2021*; *Tønnessen, Haugen & Shalfawi, 2013*; *Zhang et al., 2015*). Given its influential role in performance, reaction time becomes a

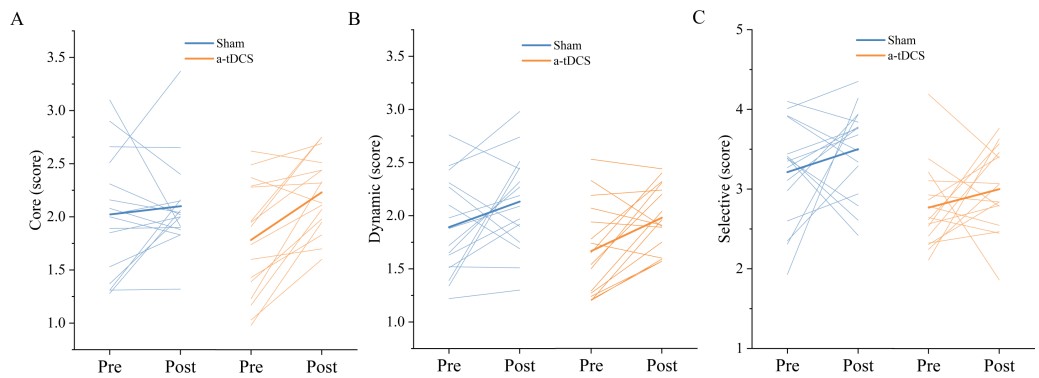

**Figure 7** **Trends of the speed threshold score of the core mode (A), dynamic mode (B) and selective mode (C) before (pre) and after (post) stimulation in each subject after tDCS (a-tDCS and sham).** Bold lines represent mean data.

crucial determinant, thereby tempting individuals to seek methods to minimize its duration. However, our results found that fencers did not have a significant reduction in reaction time after receiving a-tDCS in the M1 area. Previous research has demonstrated that a-tDCS has the potential to decrease reaction time in individuals without any known neurological disorders. *Carlsen, Eagles & MacKinnon (2015)* conducted a study involving ten healthy adults and observed a significant reduction in response time during simple response tasks following the application of a-tDCS over the SMA. Similarly, *Molero-Chamizo et al. (2018)* replicated these findings with a sample of 60 healthy adults. Notably, when a-tDCS was administered to the M1 region, participants exhibited a significant decrease in reaction time during simple response tasks. Additionally, *Bender, Filmer & Dux (2017)* found that a-tDCS over the pre-SMA significantly reduced the reaction time of 18 healthy adults in the SST test. However, a study conducted by Seidel and Ragert yielded similar findings to our own study (*Seidel & Ragert, 2019*), wherein 25 football and handball athletes who underwent a-tDCS stimulation in the M1 area did not exhibit any noteworthy enhancement in their reaction time during simple response task assessments. Despite the utilization of uncomplicated reaction time tasks to assess reaction time in these studies, divergent results were observed between healthy individuals and athletes, implying that the effects may be contingent upon the sensitivity of the test task. Moreover, there exists evidence indicating that choice reaction time tasks exhibit greater sensitivity towards the advantages of tDCS (*Hupfeld et al., 2017*). However, the current study involved the execution of a choice reaction time task, yet the observed reaction time did not exhibit a decrease. This outcome could potentially be attributed to the considerable variability inherent in athletes' neural functioning, as well as the limited efficacy of the stimulation itself. Based on the ''neural efficiency'' hypothesis and the principle of ''homeostatic plasticity'', it is posited that athletes' cerebral activity is characterized by reduced neural resource utilization, thereby achieving a state of equilibrium and efficiency during task execution (*Dunst et al., 2014*; *Ragert et al., 2009*). Consequently, the introduction of supplementary stimulation may

disturb this equilibrium, leading to inhibition of cortical excitability or ineffective effects at the behavioral level.

Response inhibition is a crucial aspect of cognitive and behavioral control, particularly in the context of professional athletes. It pertains to an athlete's capacity to suppress impulsive reactions when confronted with deceptive maneuvers from opponents or deliberate displays of tactical intentions, thereby exerting a significant influence on the outcome of a match (*Zhang et al., 2015*). In this study, we found that SSRT was significantly lower in fencers after 20 min of 2 mA a-tDCS in the M1 area, suggesting that response inhibition is enhanced after a-tDCS. Our study yielded similar findings to previous studies in the field, indicating a notable reduction in SSRT following a-tDCS. Furthermore, the stimulated brain regions encompassed a broad distribution across pre-SMA, rIFG, and DLPFC, implying that response inhibition is influenced by a multitude of brain regions (*Friehs et al., 2021*; *Friehs and Frings, 2018*; *Fujiyama et al., 2022*; *Kwon & Kwon, 2013a*; *Kwon & Kwon, 2013b*; *Li et al., 2019*). In light of the aforementioned findings, two studies have elucidated potential mechanisms through which a-tDCS could augment response inhibition. *Liang et al. (2014)* conducted a study that confirmed the significant impact of a-tDCS on both SSRT reduction and electroencephalogram's (EEG) multiscale entropy (MSE) increase. Their findings indicated a positive association between improved response inhibition and higher MSE. These results were further supported by *Yu et al. (2015)*, who observed a significant decrease in SSRT following a-tDCS compared to baseline or sham stimulation. Additionally, *Yu et al. (2015)* found a notable increase in blood oxygen levels in the pre-SMA and the ventral medial prefrontal cortex (vmPFC), suggesting enhanced functional connectivity. *Muthalib et al. (2018)* demonstrated an increase in cerebral oxygenation within the 4 × 1 electrode montage covering the stimulating M1, suggesting a hemodynamic correlate of the electrical field distribution. A possible explanation is that a-tDCS across M1 increases cerebral oxygenation, generating excitatory signals that modulated motor output and biased the competition between motor responses (*Hsu et al., 2011*; *Stinear, Coxon & Byblow, 2009*). Connectivity between M1 and subcortical basal ganglia and subthalamic nuclei has been shown to play an important role in facilitating and enhancing selectivity of desired motor output, while inhibiting or suppressing undesired motor output (*Alexander, De Long & Strick, 1986*; *Nambu, Tokuno & Takada, 2002*). Furthermore, there is evidence of a functional link between M1 and the forebrain-striatum-thalamus network activated by behavioral inhibition (*Aron, 2007*; *Aron & Poldrack, 2006*; *Neubert et al., 2010*). Stimulation of M1 may activate the inhibitory function of the forebrain-striatum-thalamus network thereby enhancing response inhibition. However, further in-depth studies of neurophysiological and functional brain imaging are needed.

The concept of attention serves as a fundamental element and intermediary among various facets of cognitive functioning, exhibiting a strong association with perception, memory, and other cognitive capacities (*Dijksterhuis and Aarts, 2010*; *Vaughan & Laborde, 2021*). Moreover, attention exerts a significant influence on athletes' aptitude in acquiring specific sports skills, while also conferring a noteworthy psychological advantage in attaining competitive prowess, thereby assuming a crucial role in determining competition outcomes (*Vaughan & Laborde, 2021*). The current study observed a significant improvement in

fencers' attentional allocation (core mode) following the a-tDCS intervention, aligning with prior research findings. Previous studies have demonstrated that a-tDCS not only enhances attention in patients but also in healthy individuals (*Fukai et al., 2019*; *Lu et al., 2020*; *Nejati et al., 2021*; *Fazeli et al., 2019*; *Silva et al., 2017*), indicating its potential for valuable application in rehabilitation to restore attention and facilitate reintegration into daily life. Furthermore, a-tDCS may also contribute to heightened attention and improved efficiency in memory-learning among the general population. Moreover, enhancing attention allocation can potentially facilitate fencers in promptly evaluating the match scenario, capitalizing on advantageous positions and opportunities, and adapting the tempo and strategies of the match. This is advantageous for attaining favorable match performance, as evidenced by a previous investigation that established a noteworthy association between attention and fencing performance (*Hijazi, 2013*). However, we do not know if this type of stimulation will have any effect on fencing performance itself.

The precise mechanisms underlying the attention-enhancing effects of a-tDCS remain elusive. However, it is well-established that attentional processing is a multifaceted phenomenon. Notably, functional magnetic resonance imaging studies have identified key brain regions involved in attentional processing, including the prefrontal oculomotor area, prefrontal lobe, inferior parietal cortex, and intraparietal sulcus, collectively forming the frontoparietal network of attention (*Dixon et al., 2017*). Furthermore, previous research conducted by *Nelson et al. (2014)* has indicated that a-tDCS leads to an acceleration of cerebral blood flow and an enhancement of blood oxygen saturation. This finding suggests that the augmentation of attention through a-tDCS may be attributed to these physiological changes. Consequently, it is plausible to propose that the application of a-tDCS across M1 for the purpose of enhancing attention could be accountable for these effects. Moreover, prior research integrating functional magnetic resonance imaging (fMRI) and tDCS in a sequential or concurrent manner has yielded empirical support indicating that the modulatory impacts of tDCS extend beyond the targeted region, exerting influence on task-related neural activity and connectivity across distributed regions, encompassing both neighboring areas and distant network nodes (*Fiori et al., 2018*; *Meinzer et al., 2013*; *Weber et al., 2014*; *Zheng, Alsop & Schlaug, 2011*). The a-tDCS to M1 may promote activity in brain regions important for attentional processing, such as prefrontal eye movement areas, prefrontal lobe, inferior parietal cortex, and intraparietal sulcus, thereby enhancing attention. However, further explorations are required in the future.

Our study did not reveal any statistically significant augmentation in dynamic attention (dynamic mode) and selective attention (selective mode). Nonetheless, it is crucial to acknowledge the significance of dynamic attention in terms of the capacity to perceive and discern alterations in the trajectory or movement pattern of multiple targets within the visual field (*Brockhoff & Huff, 2016*). Similarly, the importance of selective attention (selective mode) lies in the ability to allocate high-quality attention to the most crucial targets while avoiding wasteful allocation of attention (*Parsons et al., 2016*), as emphasized in fencing. In a particular study, a noteworthy enhancement in selective attention was observed in the M1 region of 14 boxers subsequent to a-tDCS (*Kamali et al., 2021*). However, it is worth noting that the corticospinal cord was concurrently stimulated in

that study, potentially yielding dissimilar outcomes compared to the current investigation. Additionally, disparities in the selective attention task and stimulation duration employed in the experiment may have contributed to incongruous findings as well. Furthermore, the utilization of comparable attention tasks in this study may have resulted in a certain degree of learning, potentially contributing to the absence of a substantial enhancement in dynamic attention and choice attention. In addition to the aforementioned considerations, it is crucial to acknowledge that variations in subject populations, as well as the substantial intra-subject differences, can yield disparate outcomes. Consequently, it is advisable for forthcoming investigations to standardize both the stimulation paradigm and the test task, thereby mitigating heterogeneity and fostering more pertinent research pertaining to the athlete population.

This study has some limitations. First, all included subjects were female fencers; therefore, whether the conclusions drawn can be generalized to both sexes fencers need further validation. Second, we only investigated the effects of a-tDCS on reaction time, response inhibition and attention after a single session of a-tDCS and hence, the long-term effects of this therapy need further investigation. Third, we did not ask subjects after the experiment if they suspected they received actual stimulation in one condition and sham stimulation in another, the validity of blinding was not assessed in this study and should be evaluated in future studies. Fourth, the sample size used is still not large enough and needs to be increased in the future. Finally, neuroimaging techniques were not used to investigate the underlying mechanisms of tDCS effects, and future studies should incorporate these techniques to investigate possible mechanisms.

## CONCLUSIONS

A single session of anodal tDCS could improve response inhibition, attention allocation in female fencers. This shows that tDCS has a greater potential to improve athletes' cognitive performance, although we do not know if such improvements would transfer to improved performance in competition. However, more studies involving all genders, large samples, and different sports groups are needed in the future to further validate the effect of tDCS in improving the cognitive performance of athletes.

## ACKNOWLEDGEMENTS

We thank all the subjects' voluntary contributions during the completion of this study. We express our gratitude to Jiale Lv, Zhekai Yue, Junyi Li and Junlong Ma for assistance with recruitment and testing of study participants.

### Funding

This study was funded by the Open subject of the Key Laboratory of the Nanjing Sports Institute (No. SYS202103) and the Jiangsu Province graduate Research Innovation Program

(No. KYCX22_2253). The funders had no role in study design, data collection and analysis, decision to publish, or preparation of the manuscript.

### Grant Disclosures

The following grant information was disclosed by the authors:

Key Laboratory of Nanjing Sports Institute: SYS202103.

Jiangsu Province graduate Research Innovation Program: KYCX22_2253.

### Competing Interests

The authors declare there are no competing interests.

### Author Contributions

- Jiansong Dai conceived and designed the experiments, analyzed the data, prepared figures and/or tables, authored or reviewed drafts of the article, and approved the final draft.
- Yang Xiao conceived and designed the experiments, performed the experiments, analyzed the data, prepared figures and/or tables, authored or reviewed drafts of the article, and approved the final draft.
- Gangrui Chen conceived and designed the experiments, performed the experiments, authored or reviewed drafts of the article, and approved the final draft.
- Zhongke Gu performed the experiments, prepared figures and/or tables, and approved the final draft.
- Kai Xu analyzed the data, authored or reviewed drafts of the article, and approved the final draft.

### Human Ethics

The following information was supplied relating to ethical approvals (*i.e.*, approving body and any reference numbers):

The Ethics Committee of the Nanjing Sports Institute (Approval no. RT-2022-06)

### Data Availability

The raw measurements are available in the Supplementary Files.

### Supplemental Information

Supplemental information for this article can be found online at http://dx.doi.org/10.7717/peerj.17288#supplemental-information.

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
