# Peer review of "Anodal transcranial direct current stimulation enhances response inhibition and attention allocation in fencers"

_PeerJ, doi:10.7717/peerj.17288_

## Round 0.1 · original submission · Major Revisions

Please address all the comments made by the reviewers.

·

Basic reporting

Major points
1. The title is too general to reflect the scope of the present study. I suggest specifying the attention aspect of the cognitive performance
2. The author should strengthen the motivation of using Neuro Tracker as one of the functional tests. The big term of attention was described for implementing the NT task while the SST task was also designed to probe the response inhibition which is a more specific aspect of attentional process. Similarly, the author should narrow down the specific hypothesis being testing with this experimental design. The more detailed hypothesis related to three modes need to be clearly described.
3. Figure legend is missing and only subtitle of each Figure was shown.
4. For Figure 5 and 6, the author needs to show the difference between pre- and post-stimulation for each subject, since simply showing the value during two stages with the same color does not reflect the trend.
5. The post-hoc statistics following the ANOVA for all the comparison pairs are missing.

Minor points
1. Line 262: p-value has typo.
2. Line 125: too detailed and unnecessary

Experimental design

no comment

Validity of the findings

In the discussion, the authors so often imply the potential benefit of applying the current tDCS paradigm in the healthy non-athlete population, while the whole conclusion of the present study was based on fencer population. Moreover, non-athletes very likely have different attentive capability than the athletes population.

·

Basic reporting

The aim of this study was to test the influence of the acute effects of anodal transcranial current stimulation (tDCS) over the primary motor area on reaction time (RT), response inhibition and attention in sixteen professional female fencers. Results showed that tDCS induced improvements in response inhibition and attentional allocation.
Below are some comments for your convenience. The introduction needs to be completed and expanded to better present the rationale of the present study. Some information is missing and sometimes unclear.

Introduction
L58. Suggest "reaction time".
L59. ".. important role in fencing". Authors should specify what role is important for 'fencing'. Are they talking about general performance, cognitive variables, others? Related to fencing activity?

L59-60. This sentence is unclear and seems fundamental to the rationale of the present study. The authors provided some background, highlighting that several components of cognitive function play a role in fencing performance. They stated here that "these aspects of cognitive function are improved by engaging in non-fencing related activities". Please develop your thoughts. Basically, RT, response inhibition and attention can be modulated by specific cognitive training.

L67-73. Authors should specify the site of tDCS used in these studies and focus only on healthy adults, not patients (L73), out of scope.
L74. "Response inhibition" or "response suppression"? Also recite which is the preliminary study mentioned above.
L75. It seems that the preliminary study (to be precise) showed positive stimulation effects induced by tDCS on the three cognitive functions described above. This means that the present study is not so new and aims more or less to replicate these initial findings. The added value needs to be made clear to the readers. Please amend accordingly.


L76. At this stage, it is not clear why the authors state that "we still have much to learn about how the effects of tDCS applied to different brain regions...".
The influence of brain regions has not been introduced. Something is missing.
L78-79. The effects of tDCS on athlete populations have been well studied in recent years (see recent review DOI: 10.1016/j.brs.2022.11.007, doi: 10.1016/j.brs.2018.12.227). Be precise about the effects of the tDCS you are targeting.

L90. This sentence is unclear: "... a wide range of brain activity that may enhance the activity of brain regions...". Please revise it accordingly.
L93. Stimulation of M1 is not so novel for altering RT (see L81). Please clarify your ideas.
L96. This part of the sentence must be removed from such a scientific manuscript.
L99. "Sports performance". It (fencing performance in terms of what?) needs to be clearly defined beforehand.
Authors should report that the acute effects were investigated in the studies presented in the Introduction.

Methods
L132. The dose of 2 mA for 20 min should be reported.
L231. What is the effect size for the main / interaction factors with partial eta squared; please add this to L248 and L250, then L256 and 258, L262.

Results
L251. Keep or remove either Table 1 or Figure 5 as they are redundant in some way. The effect size can be added in the text.

Experimental design

The experimental design is appropriate and the methods are described in sufficient detail.

L132. The dosage of 2 mA for 20 min when using anodal tDCS should be referred to. Please justify this choice.

Validity of the findings

Discussion
This section is well written and relevant, proposing an in-depth discussion of the findings. I enjoyed reading it. This section is adequately organised by reporting the main findings of the three cognitive functions assessed after the stimulation conditions: RT, response inhibition and attention.
The authors could add at the beginning of this section the novelty of their study when comparing their results with those of the primary study they mentioned in the introduction (see previous comment).

L326. Authors can use the results of Muthalib et al. 2018 here. They showed an increase in cerebral oxygenation within the 4 × 1 electrode montage covering the stimulating M1, suggesting a haemodynamic correlate of the electrical field distribution.


Figure 2 is not as clear regarding the electrode position. Please improve the 3D electrode placement image by showing the anodal and cathodal electrode positions.
Figure 4 is not necessary. Please remove
Figures 5 and 6 Label * Threshold p < 0.05

Additional comments

Minor
L43. Rephrase this sentence “ …is a common // processing, is the time elapsed //.”
L51. Rephase the sentence “.., and athletes //, and …”
L55. Add a space before “It is a key..”
L146. Muthalib et al. (2018).
L282. Carlsen et al. (2015) and correct this issue in the all following references: L285, 288, 319 etc.

---

## Round 0.2 · Minor Revisions

Please address the comments from the reviewer.

·

Basic reporting

no comment

Experimental design

no comment

Validity of the findings

no comment

·

Basic reporting

The revised version of the article provided by the authors has been carefully prepared, taking into account the comments previously proposed.
The authors are requested to correct the following items (final comments):

L71, 72, 155 and 157: "neuronal excitability" instead of "neuroexcitability".

L518: "electroencephalogram's" instead of "neuroencephalogram's".

Experimental design

no comment

Validity of the findings

no comment

Additional comments

no comment

---

## Round 0.3 · accepted · Accept

All the comments were addressed by the authors. The manuscript is ready for publication.